# Communicating benign biopsy results by telephone in the NHS Breast Screening Programme: a protocol for a cluster randomised crossover trial

Sian Zena Williamson,[1] Rebecca Johnson,[2] Harbinder Kaur Sandhu,[3] Nicholas Parsons,[4] Jacquie Jenkins,[5] Margaret Casey,[6] Olive Kearins,[7] Sian Taylor-Phillips [8]

**Correspondence to**
Dr Sian Taylor-Phillips;
s.taylor-phillips@warwick.ac.uk

## ABSTRACT

**Introduction** One of the main harms from breast cancer screening is the anxiety caused by false positive results. Various factors may be associated with false-positive anxiety. One modifiable factor may be the method of communication used to deliver results. The aim of this study is to measure the effect on anxiety of receiving benign biopsy results in-person or by telephone.

**Methods and analysis** This is a multi-centre cluster randomised crossover trial in the English National Health Service Breast Screening Programme (NHSBSP) involving repeated survey measures at four time points. Participants will be women of screening age who have a biopsy following a suspicious mammography result, who ultimately receive a benign or normal (B1) result. Centres will trial both telephone and in-person results on a month-by-month basis, being randomised to which communication method will be trialled first. Women will be blinded to the method of communication they will receive. The analysis will compare women who have received telephone results and women who have received in-person results. The primary outcome measure will be anxiety (measured by the Psychological Consequences Questionnaire) after receiving results, while controlling for baseline anxiety. Secondary outcome measures will include anxiety at 3 and 6 months post-results, understanding of results and patient preferences for how results are communicated. Qualitative telephone interviews will also be conducted to further explore women's reasons for communication preferences. Qualitative and quantitative data will be integrated after initial separate analysis using the pillar integration process.

**Ethics and dissemination** This study has been approved by the Public Health England Breast Screening Programme Research Advisory Committee, (BSPRAC_0013, ODR1718_040) and the National Health Service Health Research Authority (HRA) West Midlands—Coventry & Warwickshire Research Ethics Committee (17/WM/0313). The findings from this study will be disseminated to key stakeholders within the NHSBSP and via academic publications.

**Trial registration number** ISRCTN36997684

**Trial sponsor** This research is part of a PhD award and is funded by the Economic and Social Research Council Doctoral Training Centre at the University of Warwick and

## Strengths and limitations of this study

► This is the first cluster randomised crossover trial to measure the impact of communication method on patient outcomes when delivering results in breast screening.
► The mixed-methods design adds depth to our understanding of communication preferences and the mechanisms by which anxiety may be affected.
► The study is in English breast screening centres and generalisability to other contexts should be carefully considered.

Public Health England. The sponsor for this research is Jane Prewett (sponsorship@warwick.ac.uk).

## INTRODUCTION

The UK National Health Service Breast Screening Programme (NHSBSP) is a population-based screening programme that aims to detect early signs of breast cancer. Asymptomatic women aged 50–70 are invited to attend every 3 years for a mammogram. The results of the mammogram take a maximum of 2 weeks as per NHSBSP guidelines.[1] If a suspected abnormality is found on the mammogram then the woman is invited back to the clinic for follow-up tests, usually within the next 2 weeks.

Follow-up tests can include clinical examination, a further mammogram or ultrasound. If these indicate suspicion of cancer then the woman receives a core needle biopsy, involving the removal of sections of tissue from the suspicious breast region. This tissue can then be pathologically analysed. Although the procedure is designed to be minimally invasive, leaving only minor bruising, some women find the experience painful and distressing. It is standard practice to perform all follow-up tests (clinical exam,

mammogram, ultrasound and biopsy) on the same day in an assessment clinic to avoid unnecessary extra waiting time for patients. Results of diagnostic tests are discussed on a case-by-case basis at multidisciplinary team meetings using a triple-assessment of clinical examination, imaging and biopsy report. Clinical guidelines recommend that all follow-up results should be delivered to the woman within 1 week.[2] These results are delivered by either telephone or in-person, depending on the procedure at the breast screening centre.

Screening programmes should provide benefit that outweighs both physical and psychological harm.[3 4] One of the main harms from breast screening is the anxiety caused by false positive results.[5] A false-positive result is when a woman has been identified at the screening phase as potentially having cancer, but follow-up tests have revealed no abnormalities. Receiving a false-positive result from screening is very common, with the majority of women who are recalled ultimately being given a false-positive result.[6] In England, around half of women receive the all clear results at the time of the follow-up tests. Women who are invited to be screened have no symptoms of breast cancer at the time of their initial mammogram. Telling a woman that something suspicious has been found in the mammogram and that further tests are needed can make her feel very anxious and believe that she might have cancer.[7] For some women, once results confirm the absence of cancer, anxiety declines. However, anxiety can remain elevated for much longer, lasting up to 3 years after receiving the benign result and leading into the next screening invitation.[8] This is an issue, as the NHSBSP has a duty to minimise the harm caused by screening.

There are various factors that may be associated with heightened anxiety during screening such as family history, lower education, younger age and individual differences in personality.[1 9–13] These factors are unmodifiable and tend to be focused at the level of the individual, which makes the minimisation of screening anxiety a challenge. However, it is possible that there are modifiable changes that can be made at the organisational level of screening that may minimise the impact of anxiety. One of these changes is the method of communication used to deliver results.

NHSBSP guidelines for communicating results state that telephone results 'should not be routinely offered'.[14] However, most breast screening centres in the UK deliver benign results to the woman over the telephone. Some breast care nurses remain concerned about how the communication method used to deliver results may contribute to the anxiety experienced by women attending screening.[2] Telephone results may offer advantages, eliminating the stress and costs associated with transport, parking and anxiously waiting in a clinic for results. However, telephone results eliminate the in-person encounter, meaning all communication is verbal only. Research from other areas has shown that non-verbal communication plays a key role in enhancing understanding and minimising anxiety.[15]

The communication methods used to deliver benign results in the NHSBSP have not yet been explored. Therefore, the impact of this communication on women receiving a benign result is unknown.

### Aim of the study
The aim of this study is to compare anxiety in women receiving benign biopsy results from the NHSBSP via telephone results or in-person.

## METHODS AND ANALYSIS
### Study design
The study design chosen was a multi-centre cluster randomised crossover trial. The randomisation allows for the direct comparison of study outcomes between women who received telephone results and women who received in-person results, while controlling for confounding factors such as education, age and individual differences in baseline anxiety.

Patient and public involvement from the charity Independent Cancer Patients' Voice was used to guide the design of the study to ensure the acceptability of the method and appropriateness of the participant materials.

### Participants and settings
This trial will be conducted in a breast screening centre setting. The study will take place in the UK where women are invited to attend breast screening every 3 years for digital mammography. The study will take place across four time points (see table 1) with a survey completed by participants at each stage. The study participants will be recruited from four English Breast Screening centres across different regions.

Participants will be women between the ages of 47 and 73 attending the NHSBSP for further tests following a suspicious mammogram. This includes women offered routine screening between ages 50 and 70 and those receiving extra rounds of screening between the ages of 47 and 49 or 71 and 73 as part of the UK age extension trial.[16] Women will be recruited at the assessment clinic pre-biopsy. However, only women who have received a benign (B2) or normal (B1) biopsy result will be included in the longitudinal data collection. Participants will not be included in the study if they presented symptomatically to the breast clinic, if they are not the recommended screening age, if they do not receive a biopsy, if they do not have English as a first or second language and if they do not have the capacity to consent.

### Measuring anxiety
The Psychological Consequences Questionnaire (PCQ) was selected as the most appropriate measure of anxiety in the breast screening setting[17] and was embedded in the participant surveys (see online supplementary appendix 1). The PCQ is a disease-specific measure, focusing on breast cancer specific anxiety across 12 questions on three dimensions: emotional, social and physical anxiety.

**Table 1** Time points for the study

| Time point | Sample | Survey content | Method |
|---|---|---|---|
| Time point 1—at assessment clinic | Women attending assessment follow-up clinic | Demographic information Communication preferences Baseline anxiety score (PCQ) Contact details | In-person |
| Time point 2—after receiving results | Women from time point 1 who had a normal or benign biopsy | Communication preferences Repeat anxiety score (PCQ) Measure of understanding | By post |
| Time point 3—3-month follow-up | Women from time point 1 who had a normal or benign biopsy | Repeat anxiety score (PCQ) | By post |
| Time point 4—6-month follow-up | Women from time point 1 who had a normal or benign biopsy | Repeat anxiety score (PCQ) | By post |

PCQ, Psychological Consequences Questionnaire.

The disease-specific measure avoids the contrast validity that is associated with using generic anxiety measures in the breast screening context.[18] The PCQ is widely used in the breast screening setting.[1 8 19–21]

Participants will rate their anxiety on the PCQ by judging each statement on a scale of 0 (not at all) to 3 (quite a lot of the time). Women will be asked "*Over the last week* how often have you experiences the following things because of *thoughts and feelings about breast cancer:*"

### Randomisation and blinding

Each centre will be randomised to one of two intervention arms by computer generated random numbers from the trial team base at the University of Warwick (see table 2). The arm relates to whether the first month at each centre will be telephone or in-person results. Each centre will commence with the communication method randomised in month 1 and continue to alternate between the two communication methods for the duration of the study. This approach allowed for each centre to use both methods of communication, controlling for previous experience (eg, centres who already telephone are more experienced and therefore women receiving telephone results from this centre might be less anxious).

**Table 2** Allocation of communication method by arm of the trial

| Month | Arm 1 | Arm 2 |
|---|---|---|
| 1 | Telephone | In-person |
| 2 | In-person | Telephone |
| 3 | Telephone | In-person |
| 4 | In-person | Telephone |
| 5 | Telephone | In-person |
| 6 | In-person | Telephone |

This approach was selected instead of a block approach, where each centre would deliver 6 months in-person followed by 6 months of telephone. The reason for this was to avoid potential bias from unforeseen centre drop out.

This design was chosen to ensure balance between trial arms at each site while addressing practical constraints. Balance between trial arms at each site is important to account for centre-level confounders such as staff communication skills and centre-level processes.

Individual randomisation was not possible as it interfered with screening centres workflow and fail-safe mechanisms to ensure that every woman was contacted with the correct information within the correct time frame.

Women will be allocated to receive their result based on the date of their attendance at the assessment visit. However, participants will be blinded to the randomisation month. Participants will only become aware of the communication method allocated when the results are received (see 'Allocation of communication method' section below for further detail). Breast screening staff will not be blinded, as they will be delivering the results and scheduling the appointments.

### Allocation of communication method

There are two types of centres who may be involved in the research: centres who currently deliver benign results in-person and centres who currently deliver benign results by telephone.

For centres currently delivering results in-person, the following process for scheduling a results appointment will be observed: During in-person study months, all consenting women at time point 1 will be given an appointment to re-attend to receive their results in-person and will receive the result in-person. During telephone study months, all consenting women at time point 1 will be given an appointment to re-attend to receive their results

in person. However, they will instead be telephoned prior to their scheduled appointment. Only benign women will be telephoned, with all women receiving other results (eg, cancer) attending their scheduled appointment.

For centres currently delivering results by telephone, the following process for scheduling a results appointment will be observed: All consenting women will be informed that, when their results are ready, they will be contacted by telephone to arrange an appointment to come back for their results. During the 'telephone' months of the study, women will be telephoned as expected. However, instead of arranging an appointment during this telephone call, results will be delivered. This means that, for women who go on to have a cancer result, they can be telephoned to arrange an appointment to attend in-person, as expected. This is in line with standard practice at these centres. During the in-person months, all consenting women will be informed that, when their results are ready, they will be telephoned to arrange an appointment to come back for their results. These women will be telephoned to arrange an appointment to come back in-person, and at that appointment they will receive their benign/normal result.

All women not enrolled in the study will receive their screening result based on the current standard practice at the attended centre.

## Data collection
### Time point 1
Participant recruitment at time point 1 will occur concurrently at each breast screening centre.

Women will be approached during their assessment visit by breast care nurses with good clinical practice training. These women would have been recalled from a previous mammogram and may have a biopsy as part of the assessment clinic. The study will be explained to potential participants and nurses will go through the informed consent process. Consenting women will fill out the time point 1 survey with study responses collected and stored securely before the participant leaves the assessment clinic.

### Multidisciplinary team meeting
At the local-level multidisciplinary team meeting for breast screening staff, women recruited into the study will be included in further time points if they receive a benign (B2) or normal (B1) result. A breast care nurse will compile the contact details of eligible women into a spreadsheet to be sent securely to the research team.

### Time point 2
The research team will distribute time point 2 surveys to eligible women with a pre-paid return envelope. If no response is received within a week, the research team will contact women by telephone as a reminder. A maximum of two telephone contact attempts will be made. This is to ensure anxiety and understanding are measured at the crucial post-results stage of the screening process.

As part of time point 2 survey, women will be asked if they would like to participate in further research involving an interview about their experience of receiving a screening result.

### Time point 3
The research team will distribute time point 3 surveys, 3 months after the biopsy result was received.

### Time point 4
The research team will distribute time point 4 surveys, 6 months after the biopsy result was received.

### Qualitative telephone interviews
The qualitative telephone interviews will explore why women prefer certain methods of communication. Women will be recruited from the time point 2 survey. Women who express an interest in participating will be sent further information about the interviews. If they wish to participate, women will return the consent form in the pre-paid envelope. Women will then be contacted by telephone by the research team to be interviewed.

The semi-structured telephone interviews will each take 10–20 min. Questions will encompass the woman's experience of receiving a result from screening. This will involve asking how the woman felt, whether she understood her result and an exploration of her views on different methods for communicating results. Interviews will be audio-recorded and transcribed verbatim. Data collection will cease once no further themes emerge and data saturation is reached.

### Mixed-methods integration
Using a mixed-methods approach, the quantitative preference survey data will be combined with the findings from the qualitative interviews. The quantitative data ask a binary choice question regarding women's communication preferences (telephone or in-person), while the qualitative interviews expand on this by exploring how women justify certain preferences for communication. In mixing the data, the qualitative data will be used to *expand* the understanding of the findings from the quantitative surveys.[22] Expansion provides richness and detail, expanding why and how women form communication preferences and moving understanding of preferences *beyond* what quantitative data or qualitative data in isolation can elucidate. The value added by integrating the knowledge of both *what* women prefer and *why* is in the completeness of our understanding of preferences, making the evidence that will inform the NHSBSP policy decisions comprehensive and patient-centred.

### Sample size considerations
In order to determine the sample size for a clustered randomised crossover study, a full specification of the important within-cluster (centre) between-period and within-period correlations is required.[23] There is currently no available evidence on the magnitude of these correlations in our selected setting, so rather than arbitrarily

selecting values we adopt a conservative approach and assume that the (within-centre) between-period correlations are zero and proceed to power as if the design were a cluster randomised design. Although the crossover aspect of the design is not explicitly accounted for in the sample size calculation, it is incorporated fully in the analysis of the primary study outcome (PCQ anxiety score at time point 2). The aim of the study is to be able to detect a clinically significant difference of 3 points in the PCQ (the difference between the score on one statement being 0, not at all, and 3, quite a lot of the time).

Assuming the primary outcome is approximately normally distributed, and the test is at the 5% significance level with 80% power to detect an effect of the specified size, 194 participants are required at time point 2, that is 97 participants per arm. Allowing for attrition rate between time point 1 and time point 2 due to participant withdrawal (15%) and participant eligibility (50%), a total of 457 participants will be recruited at time point 1. Participant withdrawal was calculated based on a mean response rate of 60% from previous research using postal surveys in a medical setting[24] with a loss of 15% at each time point as a conservative estimate.

In order to account for clustering due to the recruiting centre, design effect was applied that inflated the sample size. The intra-cluster (within-centre) correlation coefficient was set to be 0.01 and the number of observations within each cluster was assumed to be equal. With the sample size of 194 women, divided by the number of centres[4] and then divided by the number of interventions,[2] this led to the number of observations within a cluster to be 48.2. This gave a design effect of 1.49. Taking this into account, the sample size needed to achieve statistical significance (194 women) was multiplied by the

design effect, giving a total sample of 290 women at time point 2 when rounded up (see figure 1).

31 926 women are recalled for a biopsy each year. If half of the 31 926 biopsies come back as benign, this leaves a potential sample of 15 963. This means that, on average across 80 breast screening centres, each centre will have around 200 benign biopsy results each year. Therefore, assuming 50% participation rates of eligible women, recruitment will require four centres for 1 year.

## Outcomes and study measures (primary outcome, secondary outcomes)

### Primary outcome

The primary outcome is the PCQ anxiety score at time point 2.

A comparison in anxiety score will be made between women who receive results in-person and women who receive results over the telephone.

### Secondary outcomes

Secondary outcomes are:
► PCQ anxiety score at 3-month follow-up (time point 3).
► PCQ anxiety score at 6-month follow-up (time point 4).
► Subjective understanding of results (time point 2). Measured using a survey question designed in collaboration with NHSBSP stakeholders.
► Objective understanding of results (time point 2). Measured using a survey question designed in collaboration with NHSBSP stakeholders.
► Quantitative preferences for results communication before results (time point 1).
► Quantitative preferences for results communication after results (time point 2).
► Qualitative preferences for results communication before results (time point 1).
► Qualitative preferences for results communication after results (time point 2).

## Analysis

### Quantitative data—statistical analysis

A formal and more detailed statistical analysis plan will be developed by the trial team prior to the completion of recruitment.

### *Primary outcome*

The primary analysis will use a mixed effects linear regression model to estimate the effects of communication method on anxiety (time point 2), after adjusting for baseline anxiety (time point 1). PCQ score will be treated as a continuous variable and to be approximately normally distributed for all analyses.

The model set-up and fixed and random effects are as follows:
► Response variable—anxiety at time point 2 (PCQ).
► Baseline—anxiety at time point 1 (PCQ).

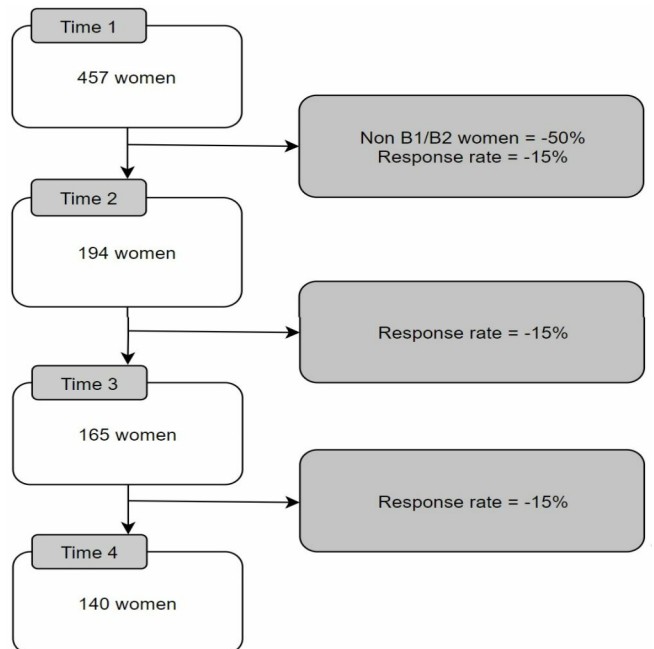

**Figure 1** Flow of required sample size for the study.

► Fixed explanatory effects (model covariates)—age, ethnicity, previous attendance, previous biopsy, education, marital status.
► Random effects—centre and temporal (period) effects.
► Comparator variable—method of communication received—telephone or in-person.

Statistical significance will be assessed at 5% level.

### Secondary outcomes

Longitudinal anxiety scores at 3 and 6 months will be analysed in the same way as the primary anxiety outcome.

Differences in understanding score between communication methods groups will be assessed using a logistic regression model, adjusting for fixed effects.

► Outcome variable—subjective understanding score (binary—yes or no), objective understanding score (binary—right or wrong).
► Fixed effects—age, ethnicity, previous attendance, previous biopsy, education, marital status.
► Comparator variable—method of communication received (telephone/in-person).

Preference data from the quantitative surveys will be presented in the form of percentages. All analyses will be implemented using IBM SPSS Statistics V.25.

### Qualitative data analysis

Qualitative preference data from time point 1 and time point 2 surveys and data from the telephone interviews will be analysed using inductive thematic analysis,[25] managed using NVivo V.10.

### Mixed-methods integration

To integrate the quantitative and qualitative preference data, the pillar integration process will be used.[26] This analytical integration technique uses four systematic stages (listing, matching, checking and pillar-building) to identify and examine connections and discrepancies in qualitative and quantitative findings. It allows for the visual display of the data and findings; this enhances overall transparency of the integration approach and the results of such an integration.

### Ethics

Attention was given to the various ethical challenges of the trial. The main ethical issue will be the use of sensitive patient information (addresses, telephone numbers). Participants will be told explicitly how their contact details will be used and stored throughout the data collection process. Participants will give informed consent for their contact details to be used for the purposes of the study.

All electronic data will be transferred securely in a password protected excel document from secure email accounts. All raw survey data will be collected directly from the centres by the lead researcher and be transferred in a secure lock-box.

### Dissemination

Results from the trial will be disseminated directly to key stakeholders within the NHSBSP. This will encourage discussion regarding how benign results are communicated in breast screening, and how this might best be implemented in order to minimise the anxiety women experience.

The results will also be disseminated via academic publications.

### Current study status

The trial began recruitment in February 2018. Data collection is due to conclude in March 2019. The trial statisticians (SW and NP) have received recruitment data but no results will be transferred to the statisticians until recruitment is closed in March 2019.

**Author affiliations**
[1]Department of Health Sciences, Warwick Medical School, University of Warwick, Coventry, UK
[2]Faculty of Health & Life Sciences, University of Warwick, Coventry, UK
[3]Clinical Trials Unit, Warwick Medical School, University of Warwick, Coventry, UK
[4]Statistics and Epidemiology, Warwick Medical School, University of Warwick, Coventry, UK
[5]National Programme Manager—NHS Breast Screening Programme, Public Health England, Sheffield, UK
[6]Clinical Nurse Specialist Breast Care, Royal Wolverhampton NHS Trust, Wolverhampton, UK
[7]National Lead Breast Screening QA, Public Health England, Birmingham, UK
[8]Population Evidence and Technologies, Warwick Medical School, University of Warwick, Coventry, UK

**Contributors** SZW is the lead researcher. SZW and ST-P drafted the manuscript. SZW, RJ, HKS and ST-P participated in the design of the study. JJ, MC and OK are key stakeholders in the NHSBSP who assisted with the study design. NP aided with the statistical analysis. All authors have reviewed and approved the manuscript.

**Funding** This research is part of a PhD award and is funded by the Economic and Social Research Council Doctoral Training Centre at the University of Warwick. The funding has been awarded for the studentship to SZW for her PhD project for 4 years of full-time study. The award consists of payment of academic fees and a maintenance award. A further contract between the University of Warwick, Public Health England and the PhD student (SZW) has secured £4000 in research expenses.

**Competing interests** None declared.

**Patient consent for publication** Not required.

**Ethics approval** This study has been approved by the Public Health England Breast Screening Programme Research Advisory Committee, (BSPRAC_0013, ODR1718_040) and the National Health Service HRA West Midlands—Coventry & Warwickshire Research Ethics Committee (17/WM/0313).

**Provenance and peer review** Not commissioned; externally peer reviewed.

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
