## [Reviewer comments · BMJ Open]

ARTICLE DETAILS

TITLE (PROVISIONAL)	Communicating benign biopsy results by telephone in the NHS Breast Screening Programme: a protocol for a cluster randomised crossover trial
AUTHORS	Williamson, Sian; Johnson, Rebecca; Sandhu, Harbinder; Parsons, Nicholas; Jenkins, Jacquie; Casey, Margaret; Kearins, Olive; Taylor-Phillips, Sian

VERSION 1 - REVIEW

REVIEWER	Emily Mulligan Winchester District Memorial Hospital Canada
REVIEW RETURNED	18-Jan-2019

GENERAL COMMENTS	I feel this is a great contribution to the field. Very thoroughly thought out and easy to follow the flow of ideas. Each section was fleshed out and well-organized to the point that I feel that this study could easily be understood and repeated. There might be some mention of the suspected limitations of this study if there are any that come straight to mind, but definitely not necessary. There are no major modifications suggested.
---

REVIEWER	Amanda Jayakody Health Behaviour Research Collaborative School of Medicine and Public Health University of Newcastle Australia
REVIEW RETURNED	27-Jan-2019

GENERAL COMMENTS	Overall a very well set out protocol however I have a few minor comments for you to address below and I've included some 'in-text' comments in the attached document: 1. Background: I feel you need to delve more into the literature. This is your study protocol and your opportunity to really justify your research objective which the background really doesn't yet. You refer to other research towards the end of the background but provide no references e.g. "Research from other areas has shown that non-verbal communication plays a key role in enhancing understanding and minimising anxiety." What other similar
--

	research studies/interventions are out there that you have learnt from? 2. PCQ: I didn't get a real sense of what the PCQ included. There may have been a typo as the description doesn't include the domains of the questions (page 8). 3. Sample size considerations: Where did you get your estimates of the withdrawal % from? Please provide a reference from a similar study as its important these estimates are based on previous work (whether a pilot or other similar screening studies)otherwise you're stabbing in the dark as it were! 4. Random assignment: I think you need a little more detail on your randomisation methods keeping in mind the Spirit/consort checklist. Who conducts randomisation? How are you randomising - you say by month but how? via a computer generated sequence? More detail needed here. 5. Qualitative interviews: how will they be conducted? Via telephone or in person? 6. Survey questions: You need some detail on what the communication preference questions will include. 7. Quant statistical analysis: what package will you use? You mention you'll use NVivo for managing the qual data. The reviewer provided a marked copy with additional comments. Please contact the publisher for full details.
--	--

REVIEWER	Sarah Liptrott European Institute of Oncology
REVIEW RETURNED	28-Jan-2019

GENERAL COMMENTS	See above
-----------

REVIEWER	Jacqueline Birks University of Oxford UK
REVIEW RETURNED	19-Mar-2019

GENERAL COMMENTS	Title : as discussed below describing the trial as a controlled trial is misleading. There is no mention that this is a cluster randomised trial. It is also a cross over trial, as there is cross over between interventions within clusters. Trial sponsor: not stated Abstract The population is not described accurately. The design is not described accurately. This study is underway, recruitment began in February 2018 and is due to conclude in March 2019. The study is described as a prospective longitudinal randomised controlled trial. The participants are women attending the NHSBSP for further tests after an initial mammogram indicated a suspected abnormality. Introduction It is important to clarify the timelines for this study and neither the background nor the methods section include sufficient detail. It is important because time between the tests may be associated with level of anxiety felt by the participants.
--

	The first time line is time between initial mammogram, initial mammogram results, and second test time. There may be guidelines on these times, which should be reported. There must be data on how centres perform which could be briefly summarised. It is not clear whether the women needing a biopsy after the second set of tests have this test on the same day as the second set of tests or whether another appointment is made. If the later what is the mean time between second set of tests and biopsy? Are data on these times being collected? Study design The study is not a controlled trial, it is a cluster randomised study comparing two methods of communicating results with cross over of method within clusters. Neither of the methods of communicating results appear to be the usual method as currently different centres choose between the two methods. Measuring anxiety How many questions are asked for a PCQ? The range of values for a PCQ is not described. Is the PCQ treated as a continuous variable for analysis? Sample size considerations This section should come after the study design has been described in detail. Randomisation and blinding How was the randomisation carried out? Women will receive their results in accordance with the month in which they attend the recall clinic. This is one reason it is important to describe the sequence of test times, whether biopsy is on the same day as the second set of tests. If not which date is used to allocate to methods of communication? Data collection Time point 1 This is the visit following the initial mammogram. Consenting women will fill out the Time point 1 survey during this visit. Describing the time of the survey as “during their follow up visit” is confusing. Analysis This section does not describe how the complex design will be analysed. Primary outcome It is not clear whether the PCQ is analysed as a continuous variable. Time between visits should be collected. It may be an important covariate. There is cross over of the methods of communication within each cluster. How is this to be dealt with in the analysis? Secondary outcomes If PCQ is analysed as a continuous variable, with the exact date of each assessment available, the best method of analysis would be a multilevel model with all assessments included to investigate how the score changed over time.
--	---

VERSION 1 – AUTHOR RESPONSE

Reviewer 1: comments and responses

Comment	Response
---------	----------

I feel this is a great contribution to the field. Very thoroughly thought out and easy to follow the flow of ideas. Each section was fleshed out and well-organized to the point that I feel that this study could easily be understood and repeated.	Thank you very much for this kind review.
There might be some mention of the suspected limitations of this study if there are any that come straight to mind, but definitely not necessary. There are no major modifications suggested.	As suggested, no limitations were added at this stage but this will be considered in future publication of the results.

Reviewer 2: comments and responses

Comment	Response
1. Background: I feel you need to delve more into the literature. This is your study protocol and your opportunity to really justify your research objective which the background really doesn't yet. You refer to other research towards the end of the background but provide no references e.g. "Research from other areas has shown that non-verbal communication plays a key role in enhancing understanding and minimising anxiety." What other similar research studies/interventions are out there that you have learnt from?	Page 5 - The final sentence of the background literature was adapted to highlight the gap in the literature and why the gap matters. Hopefully this justifies the objective in a clearer way to the audience. "The communication methods used to deliver benign results in the NHSBSP has not yet been explored. Therefore, the impact of this communication on women receiving a benign result is unknown."
2. PCQ: I didn't get a real sense of what the PCQ included. There may have been a typo as the description doesn't include the domains of the questions (page 8).	The participant survey from Time 1 has been included as an Appendix – the survey includes the PCQ measure, which provides further detail to the reader. Page 7 - Further detail has also been added in the manuscript regarding the dimensions of the PCQ measure. "The PCQ is a disease specific measure, focusing on breast cancer specific anxiety across 12 questions on three dimensions: emotional, social and physical anxiety."
3. Sample size considerations: Where did you get your estimates of the withdrawal % from? Please provide a reference from a similar study as its important these estimates are based on previous work (whether a pilot or other similar screening studies) otherwise you're stabbing in the dark as it were!	Page 13 - Further explanation has been added to explain the withdrawal percentage, which includes a reference to support this decision. "Participant withdrawal was calculated based upon a mean response rate of 60% from previous research using postal surveys in a medical setting, with a loss of 15% at each time point as a conservative estimate."
4. Random assignment: I think you need a little more detail on your randomisation methods keeping in mind the Spirit/consort checklist.	Page 8 - Further detail has been added to the randomisation methods.

Who conducts randomisation? How are you randomising - you say by month but how? via a computer generated sequence? More detail needed here.	“Each centre will be randomised to one of two intervention arms by computer generated random numbers from the University of Warwick team (see Error! Reference source not found.). The arm relates to whether the first month at each centre will be telephone or in-person results. Each centre will commence with the communication method randomised in month 1 and continue to alternative between the two communication methods for the duration of the study.”
5. Qualitative interviews: how will they be conducted? Via telephone or in person?	Clarified that this is by telephone throughout the manuscript.
6. Survey questions: You need some detail on what the communication preference questions will include.	Page 11 - Added in detail that the quantitative preference question is a binary choice. “The quantitative data asks a binary choice question regarding women’s communication preferences (telephone or in-person), whilst the qualitative interviews expand on this by exploring how women justify certain preferences for communication.”
7. Quant statistical analysis: what package will you use? You mention you’ll use NVivo for managing the qual data.	Page 15 - Added in the statistical package to be used. “Quantitative data will be analysed using IBM SPSS Statistics 25.”
In-text comment - Earlier than what? (From pg. 4)	Page 4 - This sentence has been rephrased to clarify. “The UK National Health Service Breast Screening Programme (NHSBSP) is a population-based screening programme that aims to detect early signs of breast cancer.”
In-text comment - Are there any exclusion criteria?	Page 6 - Added in some further detail to the section ‘participants and setting’.

Reviewer 3: comments and responses

n/a

Reviewer 4: comments and responses

Comment	Response
---------	----------

Title : as discussed below describing the trial as a controlled trial is misleading. There is no mention that this is a cluster randomised trial. It is also a cross over trial, as there is cross over between interventions within clusters.	The trial has been renamed to a cluster crossover trial throughout the manuscript.
Trial sponsor: not stated	Page 2 - This has been added to the abstract. “Trial sponsor – This research is part of a PhD award and is funded by the Economic and Social Research Council (ESRC) Doctoral Training Centre at the University of Warwick and Public Health England. The sponsor for this research is Jane Prewett (sponsorship@warwick.ac.uk).”
Abstract: The population is not described accurately. The design is not described accurately.	Page 2 - More detail was added to the abstract, regarding the description of the population and design. “This is a multi-centre cluster cross-over randomised controlled trial in the English NHS Breast Screening Programme (NHSBSP) involving repeated survey measures at four time points. Participants will be women of screening age who have a biopsy following a suspicious mammography result, who ultimately receive a benign or normal (B1) result. Centres will trial both telephone and in-person results on a month-by-month basis, being randomised to which communication method will be trialled first. Women will be blinded to the method of communication they will receive.”
Introduction It is important to clarify the timelines for this study and neither the background nor the methods section include sufficient detail. It is important because time between the tests may be associated with level of anxiety felt by the participants. The first time line is time between initial mammogram, initial mammogram results, and second test time. There may be guidelines on these times, which should be reported. There must be data on how centres perform which could be briefly summarised. It is not clear whether the women needing a biopsy after the second set of tests have this test on the same day as the second set of tests or whether another appointment is made. If the later what is the mean time between second set of tests and biopsy? Are data on these times being collected?	Page 4 - Clarified the time line by adding additional detail about the 2 week turn around for mammography results. “The results of the mammogram take a maximum of two weeks as per NHSBSP guidelines. If a suspected abnormality is found on the mammogram then the woman is invited back to the clinic for follow-up tests, usually within the next 2 weeks.” Page 4 - The biopsy is performed on the same day as all follow-up tests in a one stop clinic. I have clarified this in the text. “It is standard practice to perform all follow-up tests (clinical exam, mammogram, ultrasound and biopsy) on the same day in a one-stop

	assessment clinic to avoid unnecessary extra waiting time for patients. Results of diagnostic tests are discussed on a case-by-case basis at multidisciplinary team meetings (MDT) using a triple-assessment of clinical examination, imaging and biopsy report. Clinical guidelines recommend that all follow-up results should be delivered to the woman within one week. These results are delivered by either telephone or in-person, depending on the procedure at the breast screening centre.”
Study design The study is not a controlled trial, it is a cluster randomised study comparing two methods of communicating results with cross over of method within clusters. Neither of the methods of communicating results appear to be the usual method as currently different centres choose between the two methods.	The design has been clarified and named a cluster crossover randomised trial throughout.
Measuring anxiety How many questions are asked for a PCQ? The range of values for a PCQ is not described.	Page 7 - Clarified the number of questions included in the PCQ. Included the survey in the appendix for further detail. “The PCQ is a disease specific measure, focusing on breast cancer specific anxiety across 12 questions on three dimensions: emotional, social and physical anxiety.”
Sample size considerations This section should come after the study design has been described in detail.	Moved the section to later in the manuscript.
Randomisation and blinding How was the randomisation carried out? Women will receive their results in accordance with the month in which they attend the recall clinic. This is one reason it is important to describe the sequence of test times, whether biopsy is on the same day as the second set of tests. If not which date is used to allocate to methods of communication?	Page 8 - Clarified the randomisation process. “Each centre will be randomised to one of two intervention arms by computer generated random numbers from the University of Warwick team (see Error! Reference source not found.). The arm relates to whether the first month at each centre will be telephone or in-person results. Each centre will commence with the communication method randomised in month 1 and continue to alternative between the two communication methods for the duration of the study.” Added further details about the date of the one-stop follow up clinic to add clarity, which relates back to the details provided in the background.

	“Women will be allocated to receive their result based on the date of their one-stop follow-up appointment at the recall clinic.”
Data collection Time point 1 This is the visit following the initial mammogram. Consenting women will fill out the Time point 1 survey during this visit. Describing the time of the survey as “during their follow up visit” is confusing.	Page 10 - I have removed the confusing wording and hope this clarifies. “Consenting women will fill out the Time point 1 survey with study responses collected and stored securely before the participant leaves the one-stop clinic.”
Analysis This section does not describe how the complex design will be analysed.	Page 15 - Added further detail in from the Statistical Analysis Plan we have for the project. “The analysis will be based upon a Statistical Analysis Plan (SAP), designed with assistance from a statistician (NP). “PCQ score will be treated as a continuous variable for analysis. The model set-up and fixed and random effects are as follows:  • Response variable – Anxiety at time 2 (PCQ) • Baseline – Anxiety at time 1 (PCQ) • Fixed explanatory effects – age, ethnicity, previous attendance, previous biopsy, education, marital status. • Fixed sequence/period effects • Random effects – centre • Comparator variable - Method of communication received – telephone or in-person” “Differences in understanding score between communication methods groups will be assessed using a logistic regression model, adjusting for fixed effects.  • Outcome variable – Subjective understanding score (binary – yes or no), objective understanding score (binary – right or wrong) • Fixed effects – age, ethnicity, previous attendance, previous biopsy, education, marital status. • Comparator variable - Method of communication received (telephone/in-person)”
Primary outcome It is not clear whether the PCQ is analysed as a continuous variable. Time between visits should be collected. It may be an important covariate.	Unfortunately, there is no way to collect time between visits at this stage of the research, although we do appreciate that a longer build up in time between testing could certainly have an impact on anxiety.

Is the PCQ treated as a continuous variable for analysis?	Page 15 - The PCQ will be analysed as a continuous variable and this has been added. "The primary analysis will use a mixed effects model to estimate the effects of communication method on anxiety (time 2), after adjusting for baseline (time 1). PCQ score will be treated as a continuous variable for analysis."
There is cross over of the methods of communication within each cluster. How is this to be dealt with in the analysis?	Further details have been added to the analysis section (see pg. 15).
Secondary outcomes If PCQ is analysed as a continuous variable, with the exact date of each assessment available, the best method of analysis would be a multilevel model with all assessments included to investigate how the score changed over time.	Unfortunately, the time between each assessment has not been collected and is no longer possible to collect at this time. Future research could consider this approach and, in papers going forward from trial results, this is certainly a recommendation that we can make.

VERSION 2 – REVIEW

REVIEWER	Amanda Jayakody University of Newcastle, Australia
REVIEW RETURNED	20-May-2019

GENERAL COMMENTS	Thanks for addressing our feedback. I noticed one minor correction needed: Page 6, line 6 - "Study design" - The sentence here doesn't state what the study design is, but rather the governance of the design. You should move the first paragraph under "participants and settings" up to come under "Study design".
---

REVIEWER	Jacqueline Birks University Of Oxford UK
REVIEW RETURNED	09-May-2019

GENERAL COMMENTS	The authors have addressed the reviewers comments and the revised protocol includes a much clearer description of the study design. There is an outline of a statistical analysis plan. The time lines are much clearer. What is not yet clear is when the participants find out the results of the randomisation. They are clearly blinded when consenting. Do the participants know by the time they leave the clinic for the additional tests and biopsy which type of communication they will receive? If not when do they first know whether they are a telephone or in-person subject?
--

VERSION 2 – AUTHOR RESPONSE

Reviewer 4: comments and responses

Comment	Response
The authors have addressed the reviewers comments and the revised protocol includes a much clearer description of the study design. There is an outline of a statistical analysis plan. The time lines are much clearer.	Thank you for your comments. We are pleased the manuscript is now clearer.
What is not yet clear is when the participants find out the results of the randomisation. They are clearly blinded when consenting. Do the participants know by the time they leave the clinic for the additional tests and biopsy which type of communication they will receive? If not when do they first know whether they are a telephone or in-person subject?	We have added additional detail (pg. 9): “Participants will only become aware of the communication method allocated when the results are received (see ‘Allocation of communication method’ section below for further detail).” We have also clarified this allocation in the section ‘Allocation of communication method’ (pg. 9), by editing a few key words. Hopefully this clarifies the process of randomisation and the participants awareness of the method of communication used.

Reviewer 2: comments and responses

Comment	Response
Thanks for addressing our feedback. I noticed one minor correction needed: Page 6, line 6 - "Study design" - The sentence here doesn't state what the study design is, but rather the governance of the design. You should move the first paragraph under "participants and settings" up to come under "Study design".	Thank you for the feedback. We have taken the advice and moved the paragraph as recommended (pg. 6).